# Optimization of transportation routing problem for fresh food in time-varying road network: Considering both food safety reliability and temperature control

**Zhixue Zhao**[1,2]*, **Xiamiao Li**[1], **Xiancheng Zhou**[2]

**1** School of Traffic and Transportation Engineering, Central South University, Changsha, China, **2** Key Laboratory of Hunan Province for Mobile Business Intelligence, Hunan University of Technology and Business, Changsha, China

* zhaozhixue90@126.com

**Data Availability Statement:** The data used in this study are Solomon standard test data - VRPTW vehicle path. The official website of the standard case library is: (http://w.cba.neu.edu/~msolomon/

## Abstract

Study on fresh food safety reliability and temperature control has being a research focus in the fresh food cold distribution optimization study field. On this basis, optimization of transportation routing problem with time windows for fresh food in time-varying road network is studied by considering both economic cost and fresh food safety loss. A calculation method for path division strategy is designed. A food safety value loss measurement function, a metric function of energy and heat conversion a measure function of carbon emission rate are employed by considering time-varying vehicle speeds, fuel consumptions, cost of temperature control, the loss of food safety reliability and carbon emissions from transportation and temperature control. The fresh food cold chain distribution vehicle routing problem model with time windows in time-varying road network is formulated based on the objective of the distribution cost and food safety value loss minimization. According to the characteristics of the model, an adaptive improved ant colony algorithm is designed. Finally, the experimental data show that the model can effectively avoid the congestion period, reasonably control the refrigeration temperature, reduce the distribution cost, and improve food safety.

## 1 Introduction

The new economic structure has stimulated a rapid development of the fresh food e-commerce industry. People have increasingly high standards for the quality of fresh food. The current fresh food logistics distribution system is still imperfect. The high logistics costs, the serious loss and highly reduced safety of fresh food, and the low transportation rate have become the key factors in cold chain logistics that constrain the development of fresh food e-commerce. As city sizes grow, traffic congestion is frequent. Optimal planning of cold chain logistics distribution routes can not only reduce the distribution costs, ensure the hygiene and safety of fresh food, and increase customer satisfaction but also reduce energy consumption and carbon emissions, thus achieving environmental protection.

problems.htm) Because the database is public, any researcher who needs it can download it.

**Funding:** Z X C, National Natural Science Foundation of China (No. 71972069) https://isisn. nsfc.gov.cn/.

**Competing interests:** The authors have declared that no competing interests exist.

Domestic and foreign scholars have been conducting in-depth and extensive research on vehicle transportation problem. The research on vehicle route planning (VRP) [1], fuel consumption carbon emission calculation [2, 3], control mode [4, 5], safety performance analysis during driving [6], cold chain transportation etc. has become the research hotspot. A critical issue in fresh food cold chain distribution is to solve the vehicle routing problem (VRP) in fresh food distribution scientifically. With the advancement of VRP research, the fresh food cold chain distribution optimization problem has attracted the attention of more and more researchers all around the world. Osvald et al. [7] introduced a function measuring the perishability of fresh vegetables and transformed it to the costs. That study constructed a vehicle routing optimization problem model with time windows and verified its performance using Solomon's instances. Chen et al. [8] gave a full account of the perishability of fresh food. It built a nonlinear programming model considering production scheduling and distribution vehicle routing with time window constraints and solved it with a designed heuristic algorithm. The optimal production quantities, the time to start producing, and the optimal distribution route were determined. Tiwari & Peichann [9] optimized the cold chain vehicle routes with the goal of minimizing travel distance and carbon emissions and solved the problem with an improved cluster algorithm. Tarantilis et al. [10] taking meat and milk distribution in Greece as an example, studied a cold chain logistics distribution VRP with a heterogeneous fixed fleet of vehicles and multiple distribution centers and used a threshold-accepting-based algorithm to solve the problem. Amorim et al. [11] formulated a heterogeneous-fleet VRP with multiple time windows based on a food distribution problem in Portugal. Hsiao et al. [12] studied the distribution vehicle routing optimization problem of multi-type and multi-refrigerated temperature food transportation. Lan et al. [13] analyzed the influence of urban road traffic congestion on the costs of cold chain logistics vehicle distribution, constructed the corresponding vehicle routing optimization model, and designed a hybrid genetic algorithm to solve the model. Regarding uncertain demand, Li et al. [14] constructed a distribution routing optimization problem with time windows for perishable products based on stochastic customer demand. Ma et al. [15] and Shao et al. [16] solved the cold chain distribution VRP under fuzzy-stochastic demand, the conditions of uncertain customer demands and time windows, the routing optimization problem was solved for objectives regarding overall distribution cost and customer satisfaction. Kang et al. [17] built the cold chain logistics transportation models not only considered the costs of vehicle use, transportation costs, damage costs, refrigeration costs, and penalty costs but also took minimizing carbon emission costs as an optimization goal to optimize the cold chain logistics distribution routing. They used an improved ant colony algorithm, genetic algorithm, and others to solve the problem. Zhang et al. [18] designed a cold chain logistics vehicle routing mathematical model with multiple depots and multiple types of vehicles and solved it with a genetic algorithm and elite selection. In a paper on perishable fresh product distribution, Ma et al. [19] constructed an integrated optimization model of production scheduling and distribution for perishable products to maximize profits and designed a hybrid intelligent algorithm to solve it. Devapriya et al. [20] formulated an integrated production and distribution scheduling optimization model for a perishable fresh product with the objective being minimum total logistics cost. A mixed heuristic algorithm was designed to resolve the model. Wu et al. [21] studied the integrated production and distribution scheduling optimization problem with time windows of perishable food under time-varying network conditions. Wang et al. [22] focused on the deficiencies in the reasonable combination of customers' time windows and fresh product temperature control in a study on fresh product logistics distribution optimization. Considering the stringent timeliness requirements of fresh product delivery, an optimization model was constructed to minimize both the distribution logistics cost and the value loss of fresh product. From the perspectives of both

economic and environmental costs, Liu et al. [23] studied the VRP with time windows for fresh e-commerce distribution on time-varying networks. It comprehensively considered the time-varying velocity of vehicles, the perishability and spoilage of fresh agricultural products, and customers' time windows and employed the ant colony algorithm to solve the problem. Wang et al. [24] investigated optimization of Vehicle Routing Problem (VRP) with time windows for cold-chain logistics based on carbon tax in China and proposed a Cycle Evolutionary Genetic Algorithm (CEGA) to solve the model.

The current research literature on food distribution system mostly considers the physical loss generated in the process of food distribution due to perishability in VRP. However, the physical loss of food and food safety are not the same concept. Although no physical changes in some food (for example, milk, sushi) are evident, their safety could fail to meet the sanitary standards. Zwietering et al. [25] investigated a model of the influences of several temperatures on microbial growth in foods. Xie et al. [26] introduced the reliability and safety of food cold chain logistics into the distribution routing optimization problem and solved the problem with a max-min ant colony algorithm.

The results achieved so far have laid a good foundation for the research on cold chain logistics VRP. Nevertheless, limitations still exist in the following four main aspects. (1) The existing research results on cold chain logistics distribution generally assume a constant vehicle velocity, which is not consistent with real life. A few studies are based on time-varying networks. Additionally, the existing literature mostly assumes that vehicles leave the distribution center at the same time, and a few studies consider that vehicles depart at different times. (2) Most of the existing research results on cold chain logistics distribution do not consider traffic congestion, which is very common. Traffic congestion can significantly increase the driving time, vehicle energy consumption, carbon emission, and loss of food safety, seriously affecting the distribution efficiency and city environment, and thus should be studied urgently. (3) The current literature about cold chain logistics vehicle routing optimization generally only considers the physical loss of food due to perishability during fresh food distribution, while a few studies address the food safety issue, which ought to be considered in fresh food cold chain transportation distribution. (4) The current studies on fresh food logistics distribution mainly focus on optimization problems about time windows, fuel consumption, and customer satisfaction, while the temperature control-based problems associated with cold chain distribution, such as temperature control cost and fresh food safety loss, need to be further studied.

This paper first analyzes the influence of time-varying networks on vehicle routing planning under traffic congestion. The velocity of refrigerated vehicles and the effect of loading on fuel consumption and carbon emissions are considered. A fresh food safety loss function is introduced. A temperature-control-based optimization model is constructed, with the objective being minimizing the logistics costs of fresh food distribution and minimizing the fresh food safety loss. An ant colony algorithm based on a route partitioning strategy is developed to solve the model. Through calculation and sensitivity analysis, the optimal distribution routes and optimal temperatures are determined, thus achieving highly efficient cold chain logistics and environmental protection.

## 2 Problem description and model formulation

### 2.1 Problem description and basic assumptions

The distribution center, refrigerated transportation vehicles, and customer nodes constitute the fresh food distribution network. The problem is as follows. The distribution center distributes fresh food to customers. The locations, demands, and time windows of customer nodes are known. Distribution vehicles can arrive at customer nodes in advance but must wait until

the start of service time to serve. There is one type of fresh food to distribute, and its safety constantly deteriorates. The city traffic congestion periods and the congestion levels can be obtained from the transportation department. The problem is to comprehensively consider the logistics distribution costs and determine the optimal vehicle dispatching and routing plan that minimizes the total distribution costs and maximizes the food safety. To facilitate analysis and study, the following assumptions are made: (1) There is only one distribution center, which has sufficient supplies. There are many refrigerated transportation vehicles, which are of the same type. (2) Distribution vehicles depart from the cold chain logistics distribution center and return to the center immediately after completing the delivery task. (3) The demand of each customer node is less than the vehicle capacity. There is also a service time-window requirement. A vehicle can reach a client point to provide service at an earlier or later time but then will pay a certain penalty. (4) Considering the time-varying nature of network traffic, vehicle speeds in all the sections of the road network are updated every 10 min. (5) The temperature inside refrigerated vehicles are assumed to be stable and desirable during distribution. (6) Each vehicle has its own fixed cost, and time cost related to travel time in the distribution process. (7) The maximum load capacity of each vehicle is fixed. Each customer node has only one vehicle to serve it. While waiting to serve the customer, the engine is turned off to avoid fuel consumption and carbon emissions, but the temperature control equipment operates normally.

## 2.2 Symbols and variables

$N = \{0,1,\ldots n\}$: Set of all nodes in the cold chain logistics distribution network; 0 denotes the distribution center, $N^{\epsilon} = N\backslash\{0\}$ denotes the set of customers nodes

$K = \{1,2,\ldots k\}$: Set of cold chain distribution vehicles

$Q$: Maximum load capacity of vehicle (kg)

$W = \{1,2\ldots w\}$: Set of controlled temperatures $w$ (Celsius)

$T = \{T_1,T_2,\ldots,T_m\}$: Set of the time period throughout the day, $m$ is the total number of time periods.

$d_{ij}$: Driving distance (km) of the vehicle from node $i$ to node $j$, $i,j \in N$

$t_{ijk}$: Travel time (min) of the $k^{th}$ vehicle on the road $(i, j)$, $k \in K$, $i,j \in N$

$t_{ik}^{arrive}$: Time at which the $k^{th}$ vehicle arrives at point $i$

$t_{ik}^{leave}$: Time at which the $k^{th}$ vehicle leaves point $i$

$c_1$: Fixed cost of per vehicle (yuan/vehicle)

$c_2$: Time cost of vehicle per unit time(yuan/h)

$q_i$: Demand of customer node $i$, (kg), $q_i < Q$

$p$: Unit cost of transporting fresh food(yuan/kg)

$[ET_i, LT_i]$: Time window when the customer $i$ expects to be served

$t_i^s$: Unloading time needed when serving at customer node $i$(min)

$\vartheta$: Length of a sufficiently short route section in partitioning the route(km)

$F_{ijk}^R$: Travel distance(km) of the $k^{th}$ vehicle on the route section $R$ of road $(i, j)$

$v_{ijk}^R$: Travel speed(km/h) of the $k^{th}$ vehicle on the route section $R$ of road $(i, j)$

$TS_{ijk}^R$: Start time of the $k^{th}$ vehicle on the route section $R$ of road $(i, j)$

$t_{ijk}^R$: Travel time(min) of the $k^{th}$ vehicle on the route section $R$ of road $(i, j)$

$ds$: Safety influence coefficient

$W_{min}$: Temperature at which no microorganisms grow (Celsius)

$w$: Temperature of controlled temperatures (Celsius)

$COP$: Conversion rate between energy and heat

$W_o$: Ambient thermodynamic temperature outside, thermodynamic temperature (K)

$W_L$: Thermodynamic temperature of controlled temperatures $w$ (K), $W_L = 273 + w$

$f_{kw}$: Fuel consumption of unit delivery time and unit fresh product where the $k^{th}$ vehicle travels in the control temperature of $w$ (L/kg.h)

$g_{kw}$: Fuel consumption cost of unit delivery time and unit fresh product where the $k^{th}$ vehicle travels in the control temperature of $w$ (yuan/kg.h)

$\theta_w$: Cooling cost coefficient in the control temperature of $w$

$WC_w$: Total refrigeration fuel consumption cost of temperature control $w$ for all vehicles during the distribution process(yuan)

$FS_{ijkw}$: Food safety loss of the $k^{th}$ vehicle travelling from node $i$ to node $j$ at a controlled Celsius temperature $w$

$\omega_0,\omega_1,\omega_2,\omega_3,\omega_4,\omega_5,\omega_6$: Correction factors, whose values are dependent on the type of the loaded vehicle

$\chi_0, \chi_1, \chi_2, \chi_3, \chi_4, \chi_5, \chi_6, \chi_7$: Loading correction factors, whose values depend on the type of the loaded vehicle

$fc_{ijk}^R$: Fuel consumption rate of the $k^{th}$ vehicle on the route section $R$ of the road $(i, j)$, (L/km)

$FC_{ij}$: Fuel consumption cost of the vehicle on the road $(i, j)$, (yuan)

$cy_{ijk}^R$: Carbon emission of the $k^{th}$ vehicle on the route section $R$ of the road $(i, j)$, (kg/km)

$CY_{ij}$: Carbon emissions cost of the vehicle on the road $(i, j)$, (yuan)

$ce$: Carbon emissions generated by 1 L gas(kg/L)

$cf$: Unit cost of carbon emissions(yuan/kg)

$gf$: Unit cost of fuel consumption(yuan/L)

$P_e$: Cost of waiting for the unit time if the vehicle arrives at customer node in advance (yuan/min)

$P_l$: Cost of punishing for the unit time if the vehicle is late to the customer node (yuan/min)

$x_{kw}$: Dummy variable (0 and 1): $x_{kw} = 1$ if the vehicle $k$ is used at a controlled temperature of $w$, otherwise $x_{kw} = 0$

$y_{ikw}$: Dummy variable (0 and 1): $y_{ikw} = 1$ if the vehicle $k$ with a controlled temperature of $w$ serves customer node $i$, otherwise $y_{ikw} = 0$

$z_{ijkw}$: Dummy variable (0 and 1): $z_{ijkw} = 1$ if the vehicle $k$ with a controlled temperature of $w$ travels on the route $(i, j)$, otherwise $z_{ijkw} = 0$

## 2.3 Solving the driving time using route partitioning strategy

In a time-varying network, vehicle speeds vary with the time period. Therefore, travel time is difficult to calculate and needs to be rationally dealt with. Based on the previously proposed cross-time period travel time calculation methods [23, 27, 28], the real-time speed of a vehicle within a sufficiently short distance is used as its fixed speed within the time period. The following shows a step function for the speed ($v_{ij}(t)$) at various times on the road $(i, j)$:

$$v_{ij}(t) = \begin{cases} v_{ij}(T_1), t \in T_1 \\ v_{ij}(T_2), t \in T_2 \\ \cdots \\ v_{ij}(T_m), t \in T_m \end{cases} \quad i, j \in \mathbf{N} \tag{1}$$

This paper analyzes the relationships between driving time, vehicle velocity, route section, and driving distance based on the existing literature and designs a route partitioning method to solve the problem.

Step 1. Partition the route. When the $k^{th}$ vehicle runs on the route $(i, j)$, first the length of a sufficiently short route section is set to be $\vartheta$ ($\vartheta = 0.2km$), and the vehicle speed on this route section is assumed to be constant. The route $(i, j)$ is partitioned to a number of $H = \lceil d_{ij}/\vartheta \rceil$

route sections ($\lceil \rceil$ means rounding up to the nearest integer) according to $\vartheta$. The first H-1 route sections have a length of $\vartheta$, while the last section has a length of $d_{ij}-\vartheta* (H\text{-}1)$, so

$$F_{ijk}^R = \begin{cases} \vartheta, R \in \{1, 2 \ldots H - 1\} \\ d_{ij}-\vartheta* (H-1), R = H \end{cases}, d_{ij} = \sum F_{ijk}^R, \ (i, j \in \mathbf{N}, k \in \mathbf{K}) \qquad (2)$$

Step 2. Calculate the driving time on the first $H$-1 route sections. The speed of the route section is set according to the starting time of the route section within the time period. Calculate the travel time of the $k^{th}$ vehicle during the first route section on the road $(i, j)$: $TS_{ijk}^1 = t_{ik}^{leave}$, and determine the time period of $TS_{ijk}^1$. Calculate $v_{ijk}^1$ it according to formula (1), and $t_{ijk}^1 = \vartheta/v_{ijk}^1$. The time that the vehicle arrives at this route section is used as the departure time from the second route section, $TS_{ijk}^1 = TS_{ijk}^2 + t_{ijk}^1$. The driving times on the middle sections can be expressed in the same manner.

Step 3. Calculate the driving time on the last route section. $t_{ijk}^H = F_{ijk}^H/v_{ijk}^H, F_{ijk}^H = d_{ij}-\vartheta* (H-1)$.

Step 4. Calculate the total driving time of the $k^{th}$ vehicle on route $(i, j)$: $t_{ijk} = \sum_{R=1}^{H} t_{ijk}^R, \ T_{jk}^{arrive} = T_{ik}^{leave} + t_{ijk}, \ (i, \ j \in \mathbf{N}, \ k \in \mathbf{K})$. The computation terminates.

## 2.4 Food safety loss measurement function

The primary issue in the process of cold chain distribution is the hygiene and safety of food. Once a food safety problem occurs during transportation, it could induce health events and cause panic. The foodborne diseases caused by pathogenic microorganisms have become the number-one factor affecting food safety. The assumption proposed in Ref. [26], the temperature inside distribution vehicles keeps constant during the distribution process, is employed. The food safety loss $F_{ijw}$ is:

$$FS_{ijkw} = ds \cdot q_j \cdot \Delta W^2 (T_{jk}^{leave} - T_{ik}^{leave}), \quad \Delta W = \begin{cases} w - W_{\min}, w \geq W_{\min} \\ 0, \ w < W_{\min} \end{cases}, i \in \mathbf{N}, j \in \mathbf{N}' \qquad (3)$$

where $ds$ denotes the safety influence coefficient, whose value is determined by the type of food, the cold chain logistics environment, and the type of pathogenic microorganism. If $w \geq W_{\min}$, it will lead to the growth of microorganisms in food and increase the cost of food safety loss; otherwise the loss of food safety remains the same with the absence of microbial growth, but the cost of temperature control will increase significantly with the decrease of temperature. Due to the differences in customer demand, the total safety loss not only relates to the controlled temperature and the distribution time, but also depends on the customer service order.

The total safety loss cost of all vehicles during distribution $FSC_w$ can be expressed as:

$$FSC_w = P \cdot ds \cdot \Delta W^2 \sum_{k=1}^{K} \sum_{j=1}^{n} q_j (T_{jk}^{leave} - T_{0k}^{leave}) \quad \Delta W$$

$$= \begin{cases} w - W_{\min}, w \geq W_{\min} \\ 0, w < W_{\min} \end{cases}, i \in \mathbf{N}, j \in \mathbf{N}' \qquad (4)$$

## 2.5 Calculation of temperature control cost

Since different types of refrigerated food have different requirements of temperature control, the appropriate temperature needs to be set based on the characteristics of food during transportation to ensure food quality. According to the method of Ref. [22, 29], COP is used to

describe the conversion rate between energy and heat:

$$COP = \frac{W_L}{W_O - W_L} \qquad (5)$$

$W_L$ and $W_o$ are thermodynamic temperatures, corresponding to absolute temperatures, denoted as $W(K)$, and the unit is Kelvin (K). The relationship between thermodynamic temperature $W(K)$ and Celsius temperature $temp(°C)$ is: $W(K) = 273 + temp(°C)$. If the outside thermodynamic temperature $W_o$ is 303 K(outside temperature is 30°C), and the target controlled thermodynamic temperature $W_L$ is 283K($w = 10°C$), the values of COP is $COP = \frac{283}{303-283} = 14.15$. According to formula (5), COP at different temperature-controlled temperatures w can be calculated. According to the method of Ref. [22, 29], and to make the calculations more realistic, we set the benchmark: Assuming that the outside temperature is 25°C, and the cost of cooling to 10°C is one unit, i.e., $\theta_w = 1$ ($w = 10°C$). In this case, the refrigeration equipment of refrigerated vehicle adopts independent units, and the unit time fuel consumption of refrigerated equipment under full load is 2.4(L/h) which is obtained through experimental simulation, and $f_{kw} = 1.2 \times 10^{-3}$ (L/kg. h). then the cooling cost coefficient of 9°C, $\theta_w(w = 9°C)$ is 18.87/17.63 = 1.07, and the cooling cost coefficients for other controlled temperatures can be calculated. The $f_{kw}$ of different temperature control w can be calculated:

$$f_{kw} = f_{k(w=10°C)} * \theta_w = 0.0012 * \theta_w \qquad (6)$$

Table 1 lists the values of $COP$, $f_{kw}$, and $\theta_w$ at different temperatures w (°C). The cost of temperature control per unit time at different temperatures can be obtained by solving the equation:

$$g_{kw} = gf * f_{kw} \qquad (7)$$

Then, the total refrigeration fuel consumption cost $WC_w$ is:

$$WC_w = \sum_{k \in K} \sum_{j \in N'} q_j [T_{jk}^{leave} - T_{0k}^{leave}] g_{kw} \qquad (8)$$

## 2.6 Vehicle fuel consumption and carbon emission calculation

The cost of carbon emissions during cold chain distribution of food mainly includes the cost caused by fuel consumption for vehicle running and for temperature control. In this model, the carbon emissions generated by fuel consumption during transportation are mainly calculated with the methodologies to estimate emissions from transport (MEET) model [2]. The specific procedures are as follows: the carbon emission rate (g/km) of the $k^{th}$ vehicle on the

**Table 1. The values of COP, $f_{kw}$, and $\theta_w$ at different temperatures w.**

| w | COP | $\theta_w$ | $f_{kw}$ | w | COP | $\theta_w$ | $f_{kw}$ |
|---|---|---|---|---|---|---|---|
| 10 | 14.15 | 1.00 | 1.20E-03 | 2 | 11.96 | 1.58 | 1.58E-03 |
| 9 | 17.63 | 1.07 | 1.07E-03 | 1 | 11.42 | 1.65 | 1.65E-03 |
| 8 | 16.53 | 1.14 | 1.14E-03 | 0 | 10.92 | 1.73 | 1.73E-03 |
| 7 | 15.56 | 1.21 | 1.21E-03 | -1 | 10.46 | 1.80 | 1.80E-03 |
| 6 | 14.68 | 1.28 | 1.28E-03 | -2 | 10.04 | 1.88 | 1.88E-03 |
| 5 | 13.90 | 1.36 | 1.36E-03 | -3 | 9.64 | 1.96 | 1.96E-03 |
| 4 | 13.19 | 1.43 | 1.43E-03 | -4 | 9.28 | 2.03 | 2.03E-03 |
| 3 | 12.55 | 1.50 | 1.50E-03 | -5 | 8.93 | 2.11 | 2.11E-03 |

route section $R$ of road $(i,j)$ is

$$\omega(v)_{ijk}^{R} = \omega_0 + \omega_1 v + \omega_2 v^2 + \omega_3 v^3 + \omega_4/v + \omega_5/v^2 + \omega_6/v^3 \tag{9}$$

where $\omega(v)_{ijk}^{R}$ is the carbon emission rate of an empty vehicle driving on a road without slope at a speed of $v$ ($v = v_{ijk}^{R}$). The loading correction factors of the MEET model are:

$$LC_{ij} = \chi_0 + \chi_1 \gamma_{ij} + \chi_2 \gamma_{ij}^2 + \chi_3 \gamma_{ij}^3 + \chi_4 v + \chi_5 v^2 + \chi_6 v^3 + \chi_7/v \tag{10}$$

where $\gamma_{ij}$ is the ratio between the real loading and the capacity on the route $(i, j)$. Thus, the carbon emission rate of the $k^{th}$ vehicle on the route section $R$ is $cy_{ijk}^{R} = \omega(v)_{ijk}^{R} * LC_{ij}/1000$. According to ce = 2.32 [23], and the fuel consumption rate of the $k^{th}$ vehicle on the route section $R$ is $fc_{ijk}^{R} = cy_{ijk}^{R}/ce$; therefore, the carbon emissions cost of the $k^{th}$ vehicle on the route $(i, j)$ is:

$$CY_{ij} = cf \bullet \sum_{R=1}^{H} cy_{ijk}^{R} \bullet v_{ijk}^{R} \bullet t_{ijk}^{R} \tag{11}$$

The fuel consumption cost of the $k^{th}$ vehicle on the road $(i, j)$ is:

$$FC_{ij} = gf \bullet \sum_{R=1}^{H} fc_{ijk}^{R} \bullet v_{ijk}^{R} \bullet t_{ijk}^{R} \tag{12}$$

Meanwhile, the cost of carbon emissions due to temperature control during the distribution process is $\frac{WC_w}{gf} * ce * cf$.

## 2.7 Vehicle penalty cost

This penalty cost is incurred by the vehicle as a result of violating the client's time window during the distribution process. Client $i$ accepts service outside the time window, but there is a penalty cost for failing to provide service within the appointed time, and the total penalty cost $PC$ equals to

$$PC = Pe \bullet \sum_{i \in N} \sum_{k \in K} \max\{ET_i - t_{ik}^{arrive}, 0\} + Pl \bullet \sum_{i \in N} \sum_{k \in K} \max\{t_{ik}^{leave} - LT_i, 0\} \tag{13}$$

## 2.8 Mathematical model

The optimization goals of this paper are the minimum economic cost of logistics distribution (consisting of the fixed vehicle cost, time cost, time penalty cost, the cost of fuel consumption and carbon emission in transportation, the cost of fuel consumption and carbon emission in refrigeration, and cost of food safety loss). The vehicle routing problem model with time windows in time-varying road network for cold chain food distribution is as follows:

$$\min \sum_{k \in K} x_{kw} c_1 + c_2 \left( \sum_{k \in K} \sum_{i \in N} \sum_{j \in N} z_{ijkw} t_{ijk} + \sum_{i \in N^{\cdot}} y_{ikw} t_i^s \right) + PC$$

$$\sum_{k \in K} \sum_{i \in N} \sum_{j \in N} z_{ijkw} (CY_{ij} + FC_{ij}) + WC_w + \frac{WC_w}{gf} * ce * cf + FSC_w \tag{14}$$

$$\text{s.t.} \sum_{k \in K} y_{jkw} = 1, \forall j \in \mathbf{N'}, w \in \mathbf{W} \tag{15}$$

$$\sum_{i \in \mathbf{N}} z_{irkw} = \sum_{j \in \mathbf{N}} z_{rjkw} = 1, \ \forall k \in \mathbf{K}, , r \in \mathbf{N}', w \in \mathbf{W} \tag{16}$$

$$t_{ijk} = \sum_{R=1}^{H} t_{ijk}^{R}, , \forall i \in \mathbf{N}, j \in \mathbf{N}, k \in \mathbf{K} \tag{17}$$

$$\sum_{j \in \mathbf{N}'} q_j \leq Q, \ \forall k \in \mathbf{K} \tag{18}$$

$$\sum_{j \in \mathbf{N}'} z_{0jkw} \leq 1, \forall k \in \mathbf{K}, w \in \mathbf{W} \tag{19}$$

$$t_{ik}^{arrive} + t_i^s = t_{ik}^{leave}, \ \forall i \in \mathbf{N}, k \in \mathbf{K} \tag{20}$$

$$t_{jk}^{arrive} = t_{ik}^{leave} + t_{ijk}, \ \forall i,j \in \mathbf{N}, k \in \mathbf{K} \tag{21}$$

$$\sum_{j \in \mathbf{N}} z_{ijkw} = y_{ikw}, \forall k \in \mathbf{K}, i \in \mathbf{N}', w \in \mathbf{W} \tag{22}$$

$$\sum_{i \in \mathbf{N}} z_{ijkw} = y_{jkw}, \forall k \in \mathbf{K}, j \in \mathbf{N}', w \in \mathbf{W} \tag{23}$$

$$\sum_{i \in \mathbf{N}'} z_{i0kw} = \sum_{j \in \mathbf{N}'} z_{0jkw} = 1, \forall k \in \mathbf{K}, w \in \mathbf{W} \tag{24}$$

$$x_{kw} \in \{0,1\}, y_{ikw} \in \{0,1\}, z_{ijkw} \in \{0,1\} \tag{25}$$

Eq (14) is the objective function. Eq (15) means that each customer must be served exactly once. Eq (16) indicates that the vehicle arrives at and leaves from the same node. Eq (17) shows the relationship between the travel time on the whole route and the travel times on route sections. Eq (18) gives the vehicle capacity constraints. Eq (19) means that each vehicle can only be used once. Eqs (20 and 21) express the relationships between the arrival time, travel time, service time, and leaving time. Eqs (22 and 23) express the relationships between decision variables $z_{ijkw}$ and $y_{ikw}$. Eq (24) shows that all vehicles start and end at the distribution center. Eq (25) gives the variable value constraints.

## 3 The design of improved ant colony algorithm

The fresh food distribution routing optimization problem with time windows is an extension of the VRP with time window problem. The problem becomes more complex when considering traffic congestion; it is an NP-hard problem. The ant colony algorithm exhibits good characteristics, such as self-organizing capability, positive feedback mechanism, and parallel search ability, which enable it to solve complex combinatorial optimization problems more effectively and to be widely used in many fields. Based on the features of the model in this paper, an improved ant colony algorithm is designed to solve the model. The specific procedures are as follows:

Procedure 1: Initialize the relevant variables. Set the number of iterations $NC = 1$ and the maximum number of iterations as $NCmax$. Put M ants at the distribution center. Let the initial pheromone matrix. Compute the distance matrix between nodes. Set the global optimality is *bestcost*, and the current ant $m = 1$.

Procedure 2: Construct feasible solutions.

Step 1. With respect to ant $m$, put all unvisited customer nodes into the set tovisit$_m$. Set the vehicle number factors $vn = 1$;

Step 2. Calculate the transition probability that the ant $m$ moves from point $i$ to point $j$:

$$PN_{ij}^m = \begin{cases} \dfrac{[\tau_{ij}]^\alpha [\eta_{ij}]^\beta [U_{ij}]^\varphi [1/(\delta_j + 1)]^\gamma + 1}{\sum_{j \in N'} ([\tau_{ij}]^\alpha [\eta_{ij}]^\beta [U_{ij}]^\varphi [1/(\delta_j + 1)]^\gamma + 1)}, & j \in \mathbf{tovisit_m} \\ 0, & else \end{cases} \tag{26}$$

where $\tau_{ij}$ is the pheromone concentration, $\eta_{ij}$ denotes visibility $\eta_{ij} = 1/d_{ij}$; the saving matrix $U_{ij}$ is used as the prior information to guide ant's searching, where $U_{ij} = d_{i1} + d_{j1} - d_{ij}$; to address the priority when serving customer nodes, time factor $\delta_j$ is introduced,

$$\delta_j = \begin{cases} ET_i - t_{ik}^{arrive}, & t_{ik}^{arrive} < ET_i \\ 0, & t_{ik}^{arrive} \in [ET_i, LT_i] \\ t_{ik}^{arrive} - LT, & t_{ik}^{arrive} > LT_i \end{cases} \tag{27}$$

To avoid zero probability, which makes the ant unable to choose the next node, appropriate measures have been taken; $\alpha, \beta, \varphi, \gamma$ represent the relative importance of pheromone concentration, visibility, saving matrix factor, and time factor, respectively.

Step 3. Find the node $j$. In order to prevent the algorithm from falling into local optimization and premature stagnation, the strategy of state transition is improved by roulette wheel method according to Ref. [30]. The specific operation is as follows:

$$j = \begin{cases} \arg \max_{j \in \mathbf{tovisit_m}} \{P_{ij}^m\}, & \text{if } R_1 \leq R_0 \\ \text{Randomly selected point } j \ (j \in \mathbf{tovisit_m}) \text{based on roulette wheel method}, & if \ R_1 > R_0 \end{cases} \tag{28}$$

Where argmax is select the maximum operation. $R_1$ is a random number in $[0,1]$; and $R_0$ is a constant in $[0,1]$. Find the corresponding node $j$. The ant $m$ travels to customer node $j$ if $Q$ constraints are satisfied; otherwise, the ant returns to the distribution center, send out a new vehicle, $vn = vn+1$, and repeats Step 2 until all nodes have been visited.

Step 4. Compute the departure time of each vehicle as ant $m$ from the distribution center.

Step 5. $m = m+1$. Repeat Step 1 if $m < M$.

Procedure 3: Compute the objective function values of the routes. Calculate the objective function value $O(R_m)_{NC}$ of the searching route $R_m$ for each ant based on Eq (14). If $O(R_m)_{NC} \leq bestcost$, $best$cost $= O(R_m)_{NC}$. Then, $NC = NC+1$.

Procedure 4: Update global pheromones. Upgrade the pheromone values of the routes that the optimal ant has passed by.

$$\tau_{ij}(NC + 1) = \rho \cdot \tau_{ij}(NC) + \Delta\tau_{ij}^m(NC) \tag{29}$$

$$\Delta\tau_{ij}^m = \begin{cases} QC/O(R_m), & \text{Ant } m \text{ travels on the road}(i,j), \\ 0, & otherwise \end{cases} \tag{30}$$

where $\rho$ is the pheromone volatility, $\rho \in [0,1]$; $\Delta\tau_{ij}^m$ is the pheromone increment; $QC$ is a constant; $O(R_m)$ is the total distribution cost generated by this ant route search. To make sure that the pheromones accumulated on each road are not too many or too few, the pheromone concentration $\tau_{ij}$ on all routes are set within $[\tau_{\min},\tau_{\max}]$.

Procedure 5: Determine if the algorithm terminates. Check if $NC$ reaches $NCmax$. Repeat Procedure 2 if it has not; otherwise, the algorithm terminates.

## 4 Example simulation

### 4.1 Experiment setup

The data of the test example were from the international standard Solomon's VRPTW databases [2]. Considering various factors of the fresh food distribution, we used the data of the type C (concentrated distribution), type R (random distribution) and type RC (mixed distribution) examples of the international standard VRPTW database. In each example, there are 100 customer points and 1 distribution center. To be more realistic, the above cases were modified as follows: A fresh milk production enterprise distribution center needed to distribute 100 customer points within the city. The starting and ending times of the model are set to 7:00 (corresponding to time 0, i.e., 0 min) and 24:00 (corresponding to 1,020 min), respectively. The length of each time period is 10 min. According to urban traffic patterns, the period 7:30–9:00 (i.e., 4th–12th periods) and the period 17:30–19:00 (i.e., 64th–72nd periods) are set as traffic congestion periods, and the other periods are set as normal periods. According to Refs. [27, 31], let $\varpi = 0.05$. Three vehicle speeds $[VR(1-\varpi), VR(1+2\varpi), VR(1-3\varpi)]$ are allowed. Based on the $T$ value of the time period, the remainder function $\pi = \mathrm{mod}(T,3)$ is used. The $\pi$ values of 0, 1, and 2 correspond to three vehicle speeds during the normal time periods, $VR = 60(km/h)$, and that during traffic congestion hours was 30 km/h.

The algorithm was programmed with Matlab R2016a and run on a computer with 8 GB of RAM and a 1.90-GHz CPU. According to the methods in Refs. [2, 17, 23, 26, 29], the variable parameters were set as follows: The basic parameters of the vehicle are $Q = 200$ units; let one unit load be 10 kg; then the vehicle capacity is 2 tons; the correction factors in MEET model $\omega_0 = 110$, $\omega_1 = 0$, $\omega_2 = 0$, $\omega_3 = 0.000375$, $\omega_4 = 8702$, $\omega_5 = 0$, $\omega_6 = 0$; the loading correction factors in MEET model $\chi_0 = 1.27$, $\chi_1 = 0.0614$, $\chi_2 = 0$, $\chi_3 = -0.0011$, $\chi_4 = -0.00235$, $\chi_5 = 0$, $\chi_6 = 0$, $\chi_7 = -1.33$; and the model parameters are shown in Table 2, and the values of the algorithm parameters were taken from Ref. [31], as shown in Table 3.

**Table 2. Model parameters.**

| Parameter | Description | Value |
|---|---|---|
| $C_1$ | Fixed cost of per vehicle (yuan/vehicle) | 150 |
| $C_2$ | Time cost of vehicle per unit time(yuan/h) | 0.5 |
| $W_L(w)$ | Thermodynamic temperature of controlled temperatures $w$ (K) | $274K(1°C)$ |
| $W_{min}$ | Temperature at which no microorganisms grow (Celsius) | $273K(0°C)$ |
| $W_0$ | Ambient thermodynamic temperature outside, thermodynamic temperature (K) | $298K(25°C)$ |
| $gf$ | Unit cost of fuel consumption(yuan/L) | 6 |
| $cf$ | Unit cost of carbon emissions(yuan/kg) | 0.0528 |
| $P_e$ | Cost of waiting for the unit time if the vehicle arrives at customer node in advance (yuan/min) | 0.5 |
| $P_l$ | Cost of punishing for the unit time if the vehicle is late to the customer node (yuan/min) | 0.5 |
| $ds$ | Safety influence coefficient | $1*10^{-5}$ |
| $p$ | Unit cost of transporting fresh food(yuan/kg) | 10 |
| Ce | Carbon emissions generated by 1 L gas(kg) | 2.32 |

**Table 3. Ant colony algorithm parameters.**

| Parameter | Description | Value |
|---|---|---|
| M | Total ant population | 50 |
| NCmax | the maximum number of iterations | 300 |
| $\alpha$ | The coefficient of pheromone concentration | 1 |
| $\beta$ | The coefficient of visibility, | 3 |
| $\varphi$ | The coefficient of saving matrix factor, | 2 |
| $\gamma$ | The coefficient of time factor | 2 |
| QC | A constant in a pheromone update formula | 100 |
| $\rho$ | The pheromone volatility coefficient | 0.2 |
| $R_0$ | A constant in the roulette wheel method | 0.7 |

## 4.2 Experimental results

**4.2.1 Large-sized data example programming.** The example was calculated with the dataset R208. The running time was 314.7 seconds, and the overall distribution cost obtained was 4258.0 yuan. A total of 8 vehicles were dispatched for delivery. The fixed vehicle cost was 1200 yuan. The time cost was 1029.9 yuan. The time penalty cost was 714.3 yuan. The cost of fuel consumption and carbon emission in transportation was 938.60 yuan. The cost of fuel consumption and carbon emission in refrigeration was 367.9 yuan. The cost of food safety loss was 7.31yuan. The specific routing optimization plan is presented in Table 4. In Table 4, SN denotes the number of the vehicle, VR represents the driving route of the vehicle (0 denotes the distribution center and 0–100 denote customer nodes), VAT represents the time that the vehicle arrives at each customer node, and ST represents the departure time period.

It can be seen from the calculation that (1) the algorithm in this paper can yield optimal route planning in a relatively short time. (2) Combining VR and VAT, it can be seen that due to the constraints of city traffic congestion, customer service time windows, and vehicle capacity, obvious differences exist between the cold chain logistics vehicle distribution routes. Vehicle 3 and 6 served the most customer nodes, 15; Vehicle 8 served the least number of customer nodes, only 9. The reason is that the time window of each node is different from the others. Vehicle route planning must ensure a minimal cost while satisfying the customer nodes' time

**Table 4. Vehicle route optimization results of the example with the R208 dataset.**

| SN | VR | VAT | ST |
|---|---|---|---|
| 1 | 0-27-28-26-57-15-43-42-87-85-91-100-92-59- 0 | 0–5.26–21.36–47.48–101.89–124.29–140.91–160.44–178.92–201.79–215.11–227.99–244.64–256.68–286.47 | 1 |
| 2 | 0-53-58-40-21-73-72-74-22-75-56-39-67–0 | 326.97–332.23–346.87–361.72–378.14–391.87–404.74–418.46–431.79–446.64–460.98–479.22–499.58–556.28 | 33 |
| 3 | 0-89- 6-94-95-97-98-37-93-99-96-5-83-84-17-45- 0 | 0–9.47–24.10–40.43–56.75–72.75–94.75–107.58–126.53–139.10–151.73–168.65–186.97–203.07–219.47–236.80–276.96 | 1 |
| 4 | 0-52-18-60-61-16-44-14-38-86-46–8-82- 0 | 0–11.64–30.60–54.02–86.38–105.32–126.46–141.61–163.13–187.91–228.88–249.86–265.69–299.01 | 1 |
| 5 | 0-13- 2-41-23–4-55-25-54-80-68-77–3–0 | 622.14–637.50–666.36–700.78–729.36–754.07–771.83–785.63–806.85–825.34–837.69–852.96–866.28–898.47 | 63 |
| 6 | 0-69- 1-50-78-34-35-71-66-20-51–9-81-33-79-29–0 | 83.66–108.00–126.94–142.69–167.57–183.45–202.72–219.78–238.36–257.77–277.26–293.01–309.32–321.89–337.85–356.23–397.11 | 9 |
| 7 | 0-31-88-7-62-10-90-32-63-11-19-47-36–0 | 0–17.24–33.12–55.77–83.66–106.30–128.48–142.55–160.97–180.46–196.88–216.37–232.92–285.56 | 1 |
| 8 | 0-76-12-24-70-30-65-64-49-48- 0 | 195.22–210.69–227.12–253.96–301.02–315.57–354.48–412.21–435.14–464.58–503.49 | 20 |

windows. (3) From VAT and ST, it can be seen that vehicles 1, 3, 4, and 7 depart at time 0 from the distribution center to deliver, and other vehicles depart at different times. If departing at time 0, a vehicle needs to bear a relatively large cost of waiting due to early arrival. This suggests that it is necessary for logistics enterprises to consider time dependence when planning the routes. They should scientifically plan the routes based on real situations, such as the road network conditions and the customer node time windows, and comprehensively consider the costs, such as those for temperature control and fuel consumption during transportation. (4) According to VAT, Vehicle 2 and 8 completely avoided the congestion periods. Vehicles 1,3, 4, 6, and 7 only entered the congestion period of the morning rush hour but did not enter the evening rush hour. Vehicle 5 experienced the evening rush hour congestion, but not the morning one. This indicates that the method proposed in this paper can help to appropriately avoid the traffic congestion periods and increase the distribution efficiency of vehicles.

**4.2.2 Example simulation at different controlled temperatures.** Take the section 4.2.1 as an example, assuming that the temperature in refrigerated vehicles varies within a certain range, and other characteristics, such as the properties of goods and customer points, customer point service order and distribution route planning, keep unchanged. The temperature control cost and the food safety loss change during transportation, affecting the logistics distribution cost and lowering the safety and value of food. The temperature in fresh food distribution vehicles was controlled within the range from $-5°C$ to $10°C$. The sensitivity relationships between refrigerated temperature, temperature control cost (The cost of fuel consumption and carbon emission in refrigeration), safety loss cost, and logistics cost are illustrated in Figs 1 and 2.

As can be seen from Fig 1, when the refrigerated temperature gradually decreased from $10°C$, the safety loss cost gradually declined, but the temperature control cost increased accordingly. At a refrigerated temperature of $10°C$, the temperature control cost during milk distribution was minimum, but the food safety loss was maximum; on the contrary, when the refrigerated temperature went below $0°C$, the safety loss of milk was zero since no microorganisms grew, but the temperature control cost of consumption and carbon emissions increased as temperatures went down. According to Fig 2, as the refrigerated temperature decreased, the total logistics distribution cost was reduced first. At a refrigerated temperature of $1°C$, the logistics cost was the lowest. Then, as the refrigerated temperature decreased, the logistics cost increased, suggesting that $1°C$ is preferable to other refrigerated temperatures, $1°C$ is the optimal refrigerated temperature for transporting milk. Neither increasing nor decreasing the temperature can optimize the total logistics distribution cost.

**4.2.3 Optimization simulation on examples with different customer distributions.** Different types of examples were used to solve the milk cold chain transportation problem. The controlled refrigerated temperature was $1°C$. Each example was solved 10 times, and the average value was taken as the final result. Table 5 shows the experimental results, in which EX represents the type of example, TC means the total distribution cost, TMC denotes the fixed costs of the vehicles, the time cost, and the time penalty cost, WC is the fuel consumption cost generated by temperature control, FC is the fuel consumption cost during driving, CF is the carbon emission cost generated by driving and temperature control, SC is the safety loss cost, and VN is the number of vehicles. The route plan of datasets C103, R204, and RC207 are illustrated in Fig 3.

The following can be seen from Fig 3 and Table 5: (1) The example of type C had the highest total logistics distribution cost, the fixed costs of the vehicles, the time cost, and the time penalty cost, safety loss cost, temperature control cost, and number of vehicles used among all types, but its fuel consumption cost generated by driving was lower than that of other types. This is mainly because the customer nodes of the type C distribution were mainly concentrated at several regions, and the driving distances between the customer nodes were relatively

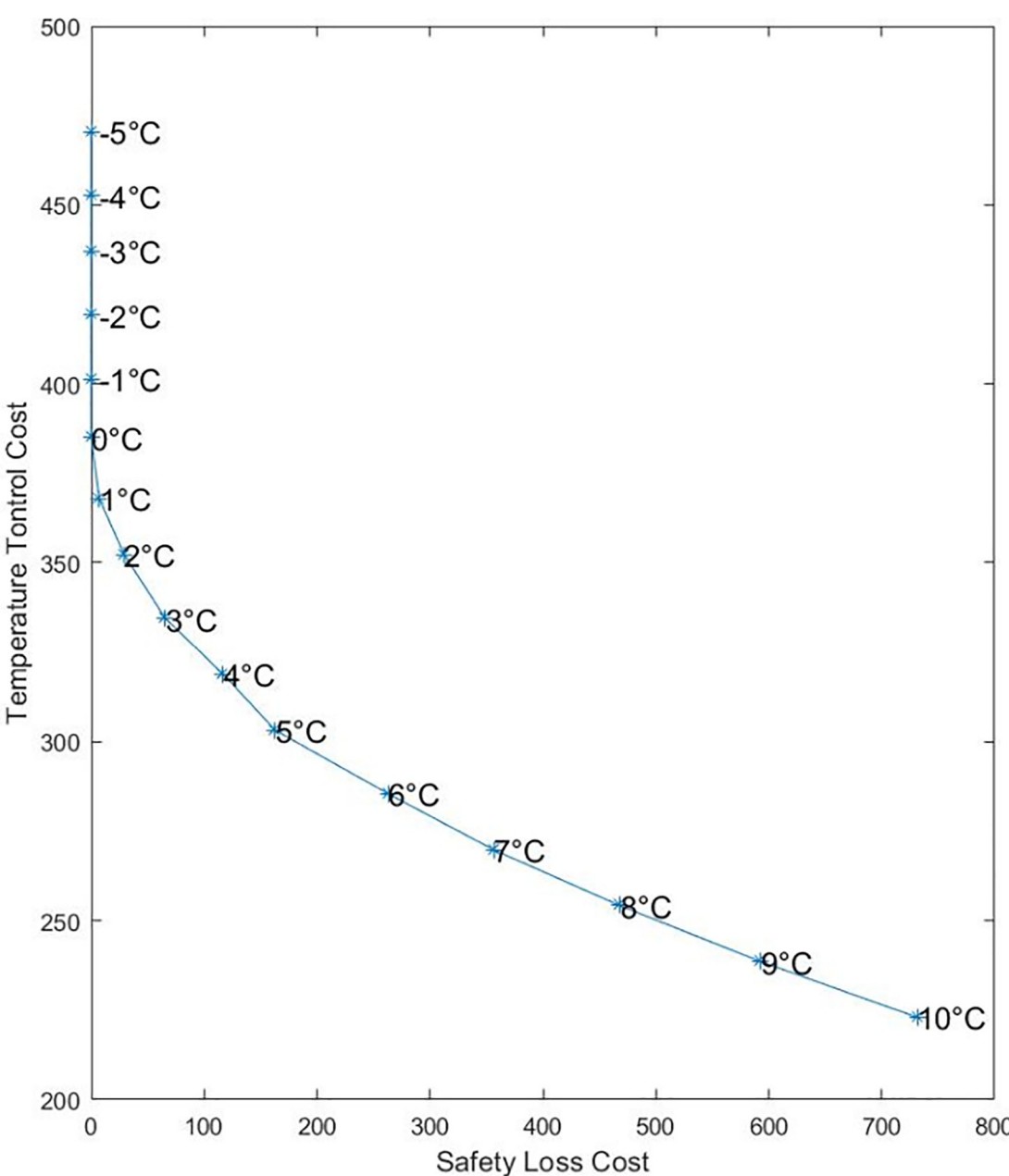

**Fig 1. The relation between safety loss and temperature control cost.**

short, which lowered the fuel consumption cost generated by driving. Nevertheless, the customers' time windows were relatively narrow. In the Solomon dataset, the service time at a customer node of the type C model was 90 minutes, as opposed to the 10-minute service time at the customer node of types R and RC. Therefore, a higher requirement was placed on vehicles to arrive at customer nodes on time and satisfy the time windows, and the time penalty cost is relatively high. At the same time, the total delivery time is the longest because the customer point service time is too long, as a result, the refrigeration time is the longest and the temperature control cost is the highest. (2) Compared to type C, types R and RC had a lower total

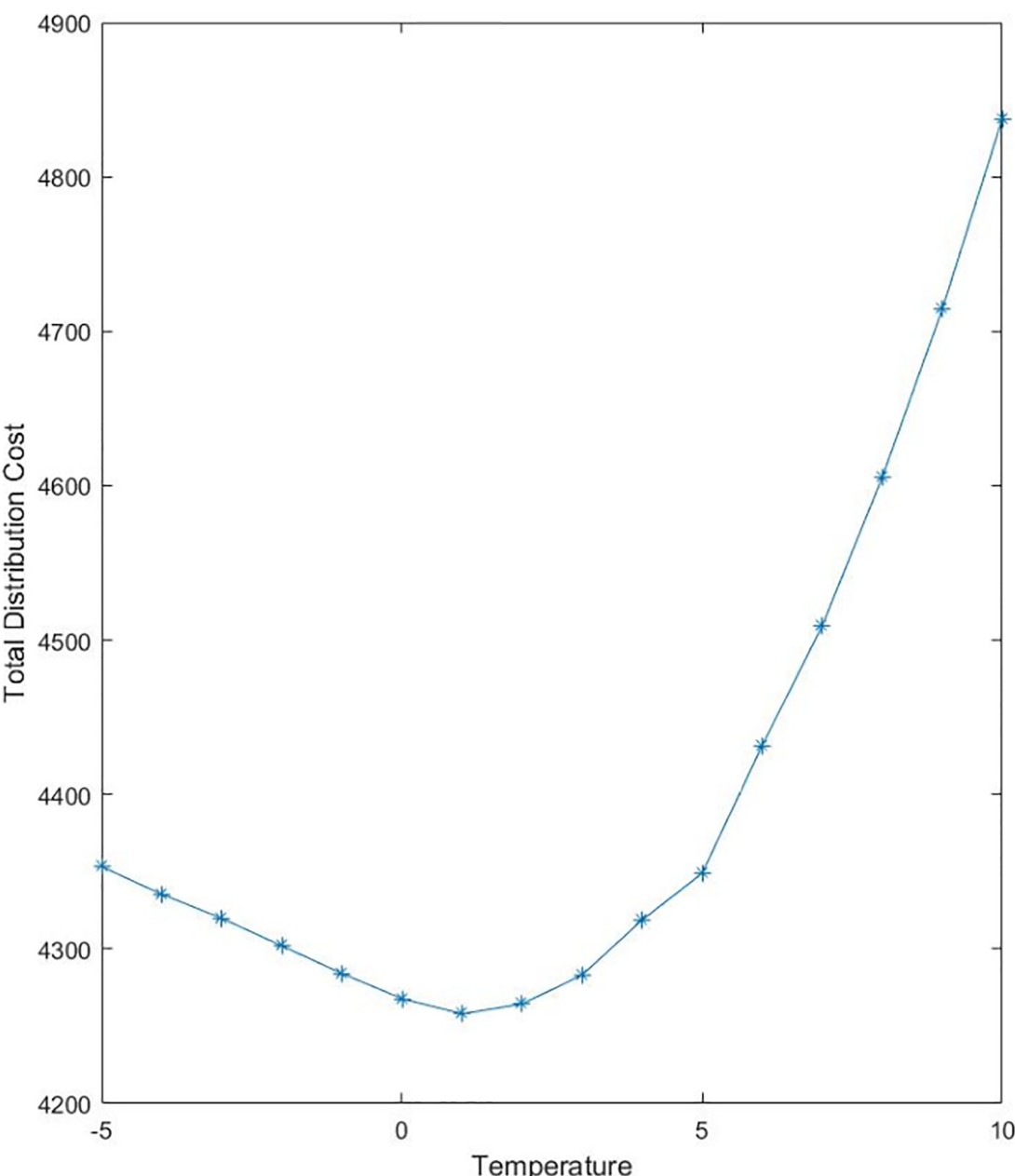

**Fig 2. Sensitivity analysis of distribution cost and refrigerating temperature.**

**Table 5. Simulation results of examples with different customer distributions.**

| EX | TC | TMC | TMC/TC | WC | FC | (WC+FC) /TC | CF | CF/TC | SC | VN |
|------|--------|--------|--------|--------|--------|-------------|------|-------|------|----|
| C103 | 11236 | 8901.8 | 79.2% | 1499.9 | 717.2 | 19.7% | 47.3 | 0.42% | 69.8 | 10 |
| C104 | 10715.8 | 8363.3 | 78.0% | 1502.5 | 732.7 | 20.9% | 47.7 | 0.45% | 69.7 | 10 |
| R203 | 5878 | 4531.3 | 77.1% | 340.6 | 946.3 | 21.9% | 26.3 | 0.45% | 33.6 | 8 |
| R204 | 4833.9 | 3554.5 | 73.5% | 335.3 | 890.3 | 25.4% | 25 | 0.52% | 28.8 | 8 |
| RC207 | 5687.2 | 4073.3 | 71.6% | 390 | 1146.3 | 27.0% | 31.4 | 0.55% | 46.3 | 9 |
| RC208 | 4598.1 | 3169.1 | 68.9% | 379.2 | 979.6 | 29.6% | 27.7 | 0.60% | 42.4 | 9 |

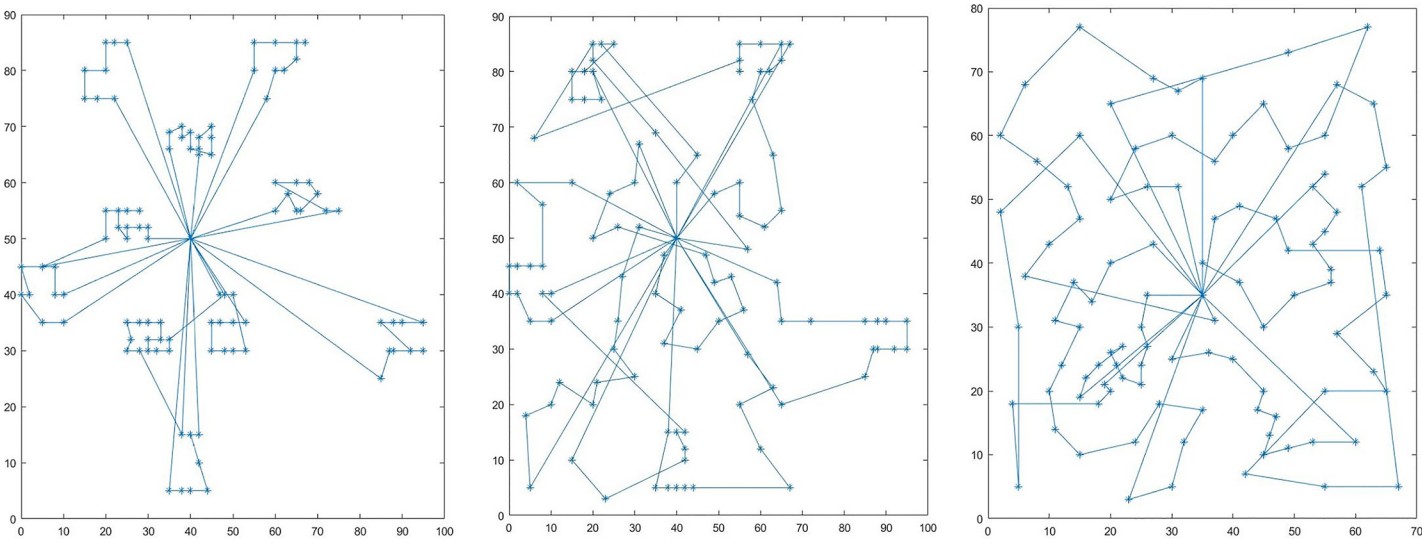

**Fig 3. Vehicle route planning for examples with different distribution types.**

distribution cost, the time cost, the time penalty cost, safety loss cost, temperature control cost, and number of vehicles. This can mainly be attributed to the 10-minute service time at the customer nodes of these two types. The customers' time window requirements were relatively loose, so the vehicles could visit many customers, and the total driving times were relatively short. At the same time, the refrigeration time is shorter and the refrigeration cost is lower. (3) For all types of examples, the TMC made up a relatively high proportion in the total distribution cost, making up approximately 70% of the total. For type C103, the proportion reached 79.2%. This indicates that the cost of vehicle usage is one of the major sources of logistics distribution cost. Since driving time is the main affecting factor, in order lower the logistics costs, the primary task should be reducing the total transportation time. (4) The fuel consumption costs generated by temperature control and driving also accounted for a relatively high proportion of the logistics costs, making up 19.7% and 20.9% of the total logistics costs for type C103 and C104. Although the fuel consumption cost during driving of type C is less, mainly because the driving distance is the least, the driving time is the longest due to the customer point service time of 90 minutes, so it needs a higher cost of temperature control fuel consumption. The total fuel consumption costs are more than 20% for types R and RC. This suggests that the fuel consumption due to temperature control and driving is also a major source of the logistics costs. Since the main affecting factors are driving time and the controlled refrigerated temperature, in order to reduce the logistics costs, the main task is to control the transportation time and properly adjust the refrigerated temperature. (5) The proportion of carbon emissions was very low, accounting for only 0.4–0.6% of the total costs, implying that from the economic perspective, the carbon emission cost caused by the current carbon tax can hardly effectively promote the logistics enterprises to save energy and reduce emissions.

**4.2.4 Comparative analysis of different algorithms.** The fresh food cold distribution optimization problem is solved by the improved ant colony algorithm (IACA), conventional ant colony algorithm (ACA) [32] and tabu search algorithm (TS) [33]. The starting time of vehicles designed in ACA and TS is 0, and there is no time factor and saving matrix in the ACA compared to IACA. We used the data of C104, R203 and RC208 to contrast experiment. Note that all of the algorithms in comparison are well tuned to achieve the best performance. The results obtained are shown in Table 6, CPUT means the computation time.

**Table 6. Calculation results solved by three algorithms.**

| EX | IACA | | ACA | | TS | |
|---|---|---|---|---|---|---|
| | **TC** | **CPUT** | **TC** | **CPUT** | **TC** | **CPUT** |
| C104 | 11236.0 | 369.1 | 11981.4 | 360.3 | 12118.7 | 351.3 |
| R203 | 6478.0 | 377.5 | 6588.2 | 360.1 | 6656.4 | 358.4 |
| RC208 | 4598.1 | 372.3 | 4732.3 | 360.2 | 4915.6 | 355.9 |

As can be seen from Table 6: (1) the distribution cost by the IACA is lower than those by the two conventional algorithms. (2) In terms of computing time, compared with the two conventional algorithms, the IACA in this paper mainly considers the starting time calculation and the improvement of state transition probability, but the time consuming of the three algorithms is similar, and all of them can get the optimal results in relatively less time. (3) Although the calculation time is relatively fast in the TS. However, it is highly dependent on the initial value, and the algorithm suffers from convergence problems.

## 5. Conclusions

Traffic congestion has become a common phenomenon in cities, which prolongs the driving time during cold chain logistics transportation and significantly increases both the temperature control cost and the food safety loss. Meanwhile, the generated carbon emissions pollute the atmosphere of cities. With respect to the fresh food cold chain distribution optimization problem under time-varying road network conditions, this paper constructs a vehicle routing problem with time windows model considering temperature control and food safety and designs a corresponding ant colony algorithm to solve the model. The experimental results suggest that (1) logistics enterprises should appropriately plan the distribution routes based on real situations, such as the city road network, customer distribution, time windows, properties of fresh food, and the outside environment, in order to effectively avoid the traffic congestion periods and reduce the total logistics costs. (2) The temperature control cost has a paradoxical relationship with the food safety loss cost. One should jointly consider the food characteristics and the outside environment to appropriately control the refrigerated temperature, thus reducing the total distribution cost. (3) The ant colony algorithm designed can effectively solve the problem in this paper, with a good convergence. (4) Currently, the carbon tax in China is relatively low, so the present carbon trading price does not notably influence the vehicle routing scheduling of logistics enterprises. Therefore, the government should properly increase the carbon tax to effectively incentivize the transportation industry to save energy and reduce emissions.

Limited by our capabilities, we only study the cold chain transportation of a single type of product with a single type of vehicle, only consider recurring traffic congestion rather than deeply analyzing the occasional traffic conditions, and only consider the linear relationship between the refrigeration fuel consumption and vehicle load for the convenience of calculation. The next step of research should focus on the actual relationship between refrigeration fuel consumption and vehicle load, the cold chain logistics transportation VRP with respect to multiple types of vehicles and products.

## Acknowledgments

We would like to acknowledge the study participants for their cooperation by sacrificing their time and proving information necessary for this study.

## Author Contributions

**Conceptualization:** Xiamiao Li, Xiancheng Zhou.

**Data curation:** Zhixue Zhao, Xiamiao Li.

**Methodology:** Zhixue Zhao, Xiancheng Zhou.

**Resources:** Zhixue Zhao.

**Software:** Xiamiao Li.

**Supervision:** Xiancheng Zhou.

**Validation:** Zhixue Zhao.

**Writing – original draft:** Zhixue Zhao, Xiamiao Li.

**Writing – review & editing:** Zhixue Zhao, Xiancheng Zhou.

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
