## [Decision Letter · Decision Letter 0]

11 May 2020

PONE-D-20-08429

Research on TDVRPTW of Fresh Food Cold Chain Distribution Optimization: Considering both Food Safety Reliability and Temperature Control

PLOS ONE

Dear Dr. Zhao,

Thank you for submitting your manuscript to PLOS ONE. After careful consideration, we feel that it has merit but does not fully meet PLOS ONE’s publication criteria as it currently stands. Therefore, we invite you to submit a revised version of the manuscript that addresses the points raised during the review process.

We would appreciate receiving your revised manuscript by Jun 25 2020 11:59PM. To enhance the reproducibility of your results, we recommend that if applicable you deposit your laboratory protocols in protocols.io, where a protocol can be assigned its own identifier (DOI) such that it can be cited independently in the future. For instructions see: http://journals.plos.org/plosone/s/submission-guidelines#loc-laboratory-protocols

We look forward to receiving your revised manuscript.

Kind regards,

Feng Chen

Academic Editor

PLOS ONE

2. Please upload a copy of Figure 4, to which you refer in your text on page 12. If the figure is no longer to be included as part of the submission please remove all reference to it within the text.

Reviewers' comments:

Reviewer's Responses to Questions

**Comments to the Author**

1. Is the manuscript technically sound, and do the data support the conclusions?

Reviewer #1: Yes

Reviewer #2: Yes

2. Has the statistical analysis been performed appropriately and rigorously? 

Reviewer #1: Yes

Reviewer #2: Yes

3. Have the authors made all data underlying the findings in their manuscript fully available?

Reviewer #1: Yes

Reviewer #2: Yes

4. Is the manuscript presented in an intelligible fashion and written in standard English?

Reviewer #1: Yes

Reviewer #2: Yes

5. Review Comments to the Author

Reviewer #1: The topic of the paper is coherent with the aims of the journal. It is very interesting, and it is attracting the attention of many researchers, as proven by recent publications on it. In particular, the paper deals with a Vehicle Routing Problem where the fresh food safety reliability and temperature control has being a research focus in the fresh food cold distribution optimization and time-varying traffic congestion is also considered. For such a problem, they propose an ant-colony optimization algorithm. But, I have several concerns that make the paper not ready to be published in this form.

(1) Some statements/symbols seem unprofessional, and nomenclature is confusing., the N={0,1,2...N} should be corrected as N={0,1,2...n} ; and the difference between a set and a function, etc.

(2) Some figure format and quality are poorly provided. For example, the Fig. 5 A.

(3) The author only considers that the vehicle arrives early at customer node. However, in reality, the vehicle may be late. That is to say, the arrive time of vehicle exceeds the customer time window upper limit.

(4) Figure 4 is missing, please check Figure no..

(5) if Δw<0 is occurs which is realistic in the section 2.3, subsequent results will result in errors, Especially in section 2.4. Furthermore, do w in the section 2.3 and TL in the section 2.4 have the same meaning?

(6) Specify the relationship between COP, gw , and θw , and the Formula gw=gf*Tfw*θw in section 2.4 is wrong.

Reviewer #2: This paper proposed a time-dependent vehicle routing problem with time windows (TDVRPTW) optimization model for cold chain food distribution. In the proposed model, the authors consider three new factors (e.g., food safety, energy consumption and carbon emission) for solving the vehicles routing problem. This paper obtains some meaningful conclusions by the case studies.

The reviewer feels that the topic has certain theoretical significance, and the paper can clearly state its contributions. However, there are some problems in this paper which make the proposed model is not right or clear. The description of parameters is confusion and inconsistent. Therefore, the reviewer suggests that this paper should be revised carefully. The details of comments are below.

1. In the section 2.2, the paper explains the relationship between the driving speed and the departure time. The travel time function is formulated. However, in the following section, the travel time function is useless because the paper partition the route into a sufficiently short section and the vehicle speed on this short section is set to be constant.

2. In the section 2.3, W_min is the temperature at which no microorganisms grow (Celsius) but it is not the minimum value of controlled temperature w. if the condition w<w_min occurs="">3. In the formula (4), t_(i-1)ik^d is not right that leads to the wrong formula (4). The formula (5) also has the same mistake.

4. In the section 2.4, the units of T_H and T_L should be clear, Celsius or Kelvin (K).

5. In table 1, the value of COP is not right if T_H=30℃.

6. Why assume θ_w=1 if T_H=30℃ and T_L=10℃?

7. In formula (5), why is the linear relationship between the customer demand and temperature control cost? As we know, the customer demand actually affects the temperature control cost. But the relationship is absolutely not linear. For example, the temperature control cost for transporting one hundred commodities is absolutely not 100 times that for transporting one commodity.

8. In formula (8) and (9), the product of v_ijkh t_ijkh^d should be equal to ϑ. Why not use ϑ instead of v_ijkh t_ijkh^d to simplify the formula (8) and (9).

9. In the section 2.7, what is the meaning of p*F in formula (10)?

10. Formula (12) is wrong. It cannot ensure that the vehicle arrives at and leaves from the same node.

11. Formula (18) and (19) cannot reflect the time window constraints properly. For example, if T_ik^arrive>ET_i, the waiting time η_ik still exists that means the vehicle has to wait for a while even it arrives within the time window.

12. In the proposed model, it lacks the constraints for y_jkw and z_ijkw. It means that if z_ijkw=1, y_jkw must be 1. Whereas, if y_jkw=1, z_ijkw can be 1 or 0.

13. In the proposed model, constraints that vehicles must return the distribution center are omitted.

14. In section 4.2.2, there is no figure 4.

15. The discussion for figure 2 is not appropriate because there is no temperature coordinate in figure 2.

16. Figure 5 should be Figure 4.

17. 5℃ is the optimal refrigerated temperature observed in figure 3. Why not set the controlled refrigerated temperature to 5℃ in the section 4.2.3.

18. In this paper, several variables are not defined before they are used (e.g., d, a, v_c, v_f, ce and etc.) and some variable definitions are duplicated (e.g., ϑ and h, k and t). A lot of variables are higher than the text and not italic. The subscripts of some variable are wrong (e.g. TL, t_(i-1)i^a ). The authors should be check all paper carefully and correct all mistakes.

 </w_min>

6. PLOS authors have the option to publish the peer review history of their article (what does this mean?). If published, this will include your full peer review and any attached files.

Reviewer #1: No

Reviewer #2: No

---

## [Author Response · Author response to Decision Letter 0]

18 Jun 2020

Reviewer#1, Concern # 1: Some statements/symbols seem unprofessional, and nomenclature is confusing., the N={0,1,2...N} should be corrected as N={0,1,2...n} ; and the difference between a set and a function, etc.

Author response: Thank you for your constructive and helpful suggestion. Some statements/symbols have been revised. All variables and sets names are standardized in section 2.2. For example, : Set of all nodes in the cold chain logistics distribution network; 0 denotes the distribution center, denotes the set of customers nodes. : Set of cold chain distribution vehicles. : Set of controlled temperatures w. : Set of the time period throughout the day, m is the total number of time periods.

In section 2 and 3, we have updated normalized statements, such as COP, , .

Reviewer#1, Concern # 2: Some figure format and quality are poorly provided. For example, the Fig. 5 A.

Author response: Thank you for your constructive and helpful suggestion. All the figures have been replaced as you required. We have uploaded the original picture to the attachment for the editorial board's review。

Reviewer#1, Concern # 3: The author only considers that the vehicle arrives early at customer node. However, In reality, the vehicle may be late. That is to say, the arrive time of vehicle exceeds the customer time window upper limit.

Author response: Thank you for your constructive and helpful suggestion. I updated the method to deal with the customer point window const raint in section 2.1: There is also a service time-window requirement. A vehicle can reach a client point to provide service at an earlier or later time but then will pay a certain penalty.

I have solved the customer point time window constraint by increasing the penalty cost in section 2.7:

2.7 Vehicle penalty cost

This penalty cost is incurred by the vehicle as a result of violating the client’s time window during the distribution process. Client i accepts service outside the time window, but there is a penalty cost for failing to provide service within the appointed time, and the total penalty cost PC equals to

 (13)

 : Time window when the customer i expects to be served 

Pe: Cost of waiting for the unit time if the vehicle arrives at customer node in advance (yuan/min)

Pl: Cost of punishing for the unit time if the vehicle is late to the customer node (yuan/min)

 : Time at which the kth vehicle arrives at point i

 : Time at which the kth vehicle leaves point i

Reviewer#1, Concern # 4: Figure 4 is missing, please check Figure no.

Author response: Thank you for your constructive and helpful suggestion. I readjusted the diagram and normalized the number.

Reviewer#1, Concern # 5: if Δw<0 is occurs which is realistic in the section 2.3, subsequent results will result in errors, Especially in section 2.4. Furthermore, do w in the section 2.3 and TL in the section 2.4 have the same meaning?

Author response: Thank you for your constructive and helpful suggestion. We have restandardized the units of T_H and T_L to in section 2.2, and we've defined the relationship between and w in section 2.5. 

w: Temperature of controlled temperatures (Celsius)

 : Ambient thermodynamic temperature outside, thermodynamic temperature (K)

 : Thermodynamic temperature of controlled temperatures w (K), 

Because Δw<0 is occurs which is realistic, so we updated the food safety loss Fijw :

 If , it will lead to the growth of microorganisms in food and increase the cost of food safety loss; otherwise the loss of food safety remains the same with the absence of microbial growth, but the cost of temperature control will increase significantly with the decrease of temperature. 

Reviewer#1, Concern # 6: Specify the relationship between COP, gw , and θw , and the Formula gw=gf*Tfw*θw in section 2.4 is wrong

Author response: Thank you for your constructive and helpful suggestion. We have identified the relationship between COP, , and in section 2.5. 

 (5);

 at different temperature control w is obtained by COP ratio;

According to the method of Ref. [22,29], and to make the calculations more realistic, we set the benchmark: Assuming that the outside temperature is , and the cost of cold storage temperature to is one unit, i.e., . In this case, the refrigeration equipment of refrigerated vehicle adopts independent units, and the unit time fuel consumption of refrigerated equipment under full load(2000kg) is 2.4(L/h) which is obtained through experimental simulation, and (L/kg.h). then the cooling cost coefficient of , (w=9 ), is 18.87/17.63=1.07, and the cooling cost coefficients for other controlled temperatures can be calculated. The of different temperature control w can be calculated:

 (6)

Table 1. The values of COP, , and at different temperatures 

w COP 

w COP 

10 14.15 1.00 1.20E-03 2 11.96 1.58 1.58E-03

9 17.63 1.07 1.07E-03 1 11.42 1.65 1.65E-03

8 16.53 1.14 1.14E-03 0 10.92 1.73 1.73E-03

7 15.56 1.21 1.21E-03 -1 10.46 1.80 1.80E-03

6 14.68 1.28 1.28E-03 -2 10.04 1.88 1.88E-03

5 13.90 1.36 1.36E-03 -3 9.64 1.96 1.96E-03

4 13.19 1.43 1.43E-03 -4 9.28 2.03 2.03E-03

3 12.55 1.50 1.50E-03 -5 8.93 2.11 2.11E-03

Table 1 lists the values of COP, , and at different temperatures w ( ). The cost of temperature control per unit time at different temperatures can be obtained by solving the equation: (7)

Reviewer#2, Concern # 1: In the section 2.2, the paper explains the relationship between the driving speed and the departure time. The travel time function is formulated. However, in the following section, the travel time function is useless because the paper partition the route into a sufficiently short section and the vehicle speed on this short section is set to be constant.

Author response: Thank you for your constructive and helpful suggestion. There are errors in the previous description, so I redescribe the change of speed under the time-varying network in section 2.3. I have modified the calculation method of driving time under time-varying network.

In a time-varying network, vehicle speeds vary with the time period. Therefore, travel time is difficult to calculate and needs to be rationally dealt with. Based on the previously proposed cross-time period travel time calculation methods [23,27,28], the real-time speed of a vehicle within a sufficiently short distance is used as its fixed speed within the time period. The following shows a step function for the speed ( ) at various times on the road (i, j):

 (1)

Reviewer#2, Concern # 2: In the section 2.3, is the temperature at which no microorganisms grow (Celsius) but it is not the minimum value of controlled temperature w. if the condition occurs, the formula (3) will get the wrong result that the food safety loss increases with the decrease of w.

Author response: Thank you for your constructive and helpful suggestion. We updated the food safety loss Fijw :

 If , it will lead to the growth of microorganisms in food and increase the cost of food safety loss; otherwise the loss of food safety remains the same with the absence of microbial growth, but the cost of temperature control will increase significantly with the decrease of temperature. 

Reviewer#2, Concern # 3: In the formula (4), t_(i-1)ik^d is not right that leads to the wrong formula (4). The formula (5) also has the same mistake.

Author response: Thank you for your constructive and helpful suggestion. We updated the formulas:

 (3) 

The driving time is calculated as: ;

Due to the differences in customer demand, the total safety loss not only relates to the controlled temperature and the distribution time but also depends on the customer service order. 

The total safety loss cost of all vehicles during distribution can be expressed as: 

 (4)

Reviewer#2, Concern # 4: In the section 2.4, the units of T_H and T_L should be clear, Celsius or Kelvin (K).

Author response: We are very grateful to the reviewee for pointing out such subtle mistake and we corrected it in the new manuscript. We restandardized the units of T_H and T_L to in section 2.2, and explained the relationship between them in the section2.5

w: Temperature of controlled temperatures (Celsius)

 : Ambient thermodynamic temperature outside, thermodynamic temperature (K)

 : Thermodynamic temperature of controlled temperatures w (K), 

Reviewer#2, Concern # 5: In table 1, the value of COP is not right if T_H=30℃.

Author response: We are very grateful to the reviewee for pointing out such subtle mistake and we corrected it in the new manuscript, and Table 1 has been modified in section 2.5.

 If the outside thermodynamic temperature is 303 K(outside temperature ) ,and the target controlled thermodynamic temperature is 283K( ), the values of COP is .

Table 1. The values of COP, , and at different temperatures 

w COP 

w COP 

10 14.15 1.00 1.20E-03 2 11.96 1.58 1.58E-03

9 17.63 1.07 1.07E-03 1 11.42 1.65 1.65E-03

8 16.53 1.14 1.14E-03 0 10.92 1.73 1.73E-03

7 15.56 1.21 1.21E-03 -1 10.46 1.80 1.80E-03

6 14.68 1.28 1.28E-03 -2 10.04 1.88 1.88E-03

5 13.90 1.36 1.36E-03 -3 9.64 1.96 1.96E-03

4 13.19 1.43 1.43E-03 -4 9.28 2.03 2.03E-03

3 12.55 1.50 1.50E-03 -5 8.93 2.11 2.11E-03

Reviewer#2, Concern # 6: Why assume θ_w=1 if T_H=30℃ and T_L=10℃?

Author response: Thank you for your constructive and helpful suggestion. 

We mainly refer to literatures[22,29] to find the benchmark of temperature control cost——When the outside thermodynamic temperature is 303 K(outside temperature ) ,and the target controlled thermodynamic temperature is 283K( ), the temperature control cost is the standard, and the cooling cost coefficient in the control temperature of w . At this case, we can we estimate that the unit time fuel consumption of refrigerated equipment under full load(2000kg) is 2.4(L/h) which is obtained by looking up relevant data on the Internet, and we can get the fuel consumption of unit delivery time and unit fresh product where the kth vehicle travels in the control temperature of : (L/kg.h). According to this standard, the cost under other temperature control can be calculated. Table 1 lists the values of COP, , and at different temperatures w ( ).

[22] Wang Y, Zhang J. Optimization Method Study of Fresh Good Logistics Distribution Based on Time Window and Temperature Control . Control and Decision [Online]. Available: DOI: 10.13195/j.kzyjc.2018.1662

[29] Wang J, Liu H T, Zhao R. The optimization of cold chain operation based on fresh food safety[J]. Systems Engineering-Theory & Practice, 2018, 1:122-134

Reviewer#2, Concern # 7: In formula (5), why is the linear relationship between the customer demand and temperature control cost? As we know, the customer demand actually affects the temperature control cost. But the relationship is absolutely not linear. For example, the temperature control cost for transporting one hundred commodities is absolutely not 100 times that for transporting one commodity.

Author response: Thank you for your constructive and helpful suggestion. We mainly refer to the calculation method of temperature control cost in literature [16]. For the convenience of calculation, there is a linear relationship between temperature control cost and weight of demand. Under the condition of fixed temperature control, the larger the weight of refrigerated fresh products, the higher the fuel consumption required for temperature control and the higher the cost. Literature [16] explains the operation steps in detail. I also modified it accordingly in section 2.5:

According to formula (5), COP at different temperature-controlled temperatures w can be calculated. According to the method of Ref. [22,29], and to make the calculations more realistic, we set the benchmark: Assuming that the outside temperature is 25 , and the cost of cooling to 10 is one unit, i.e., =1 (w=10 ). In this case, the refrigeration equipment of refrigerated vehicle adopts independent units, and the unit time fuel consumption of refrigerated equipment under full load is 2.4(L/h) which is obtained through experimental simulation, and (L/kg.h). then the cooling cost coefficient of 9 , (w=9 ), is 18.87/17.63=1.07, and the cooling cost coefficients for other controlled temperatures can be calculated. The of different temperature control w can be calculated:

 (6)

Table 1 lists the values of COP, , and at different temperatures w ( ). The cost of temperature control per unit time at different temperatures can be obtained by solving the equation:

 (7)

As you have described, the linear relationship between customer demand and temperature control cost may not be in line with the reality. To simulate their real relationship is also my research direction in the future, and I put this work in the conclusion part.

[22] Wang Y, Zhang J. Optimization Method Study of Fresh Good Logistics Distribution Based on Time Window and Temperature Control. Control and Decision [Online]. Available: DOI: 10.13195/j.kzyjc.2018.1662

Reviewer#2, Concern # 8: In formula (8) and (9), the product of v_ijkh t_ijkh^d should be equal to ϑ. Why not use ϑ instead of v_ijkh t_ijkh^d to simplify the formula (8) and (9).

Author response: Thank you for your constructive and helpful suggestion. Not all road lengths are integer multiples of ϑ. We design a route partitioning method in section 2.3: The route (i, j) is partitioned to a number of route sections ( means rounding up to the nearest integer) according to . The first H-1 route sections have a length of , while the last section has a length of . Because not all the route sections length are , the last section might be less than , the paper doesn’t use ϑ instead of v_ijkh t_ijkh^d to simplify the formula (8) and (9).

Since all road lengths are not equal, we use to represent the length of route section. At the same, due to the and of each route section is different, so the carbon emission rate and carbon emission of each route section are correspondingly different, and it is reasonable that use .

Reviewer#2, Concern # 9: In the section 2.7, what is the meaning of p*F in formula (10)?

Author response: Thank you for your constructive and helpful suggestion. p*F is the cost of safety loss. Further explanation: The total loss cost is the product of the unit price of goods and the amount of loss. To illustrate further, we have modified the total cost of the security loss FS in section 2.4:

 (4)

I also have modified the target function accordingly (14):

 (14)

Reviewer#2, Concern # 10: Formula (12) is wrong. It cannot ensure that the vehicle arrives at and leaves from the same node.

Author response: Thank you for your constructive and helpful suggestion. I have modified the formula (12) to formula (16):

 （16）

Formula (16) indicates that the vehicle arrives at and leaves from the same node.

Reviewer#2, Concern # 11: Formula (18) and (19) cannot reflect the time window constraints properly. For example, if T_ik^arrive>ET_i, the waiting time η_ik still exists that means the vehicle has to wait for a while even it arrives within the time window.

Author response: Thank you for your constructive and helpful suggestion. I updated the method the customer point window constraint in section 2.1: There is also a service time-window requirement. A vehicle can reach a client point to provide service at an earlier or later time but then will pay a certain penalty.

I have solved the customer point time window constraint by increasing the penalty cost in section 2.7:

2.7 Vehicle penalty cost

This penalty cost is incurred by the vehicle as a result of violating the client’s time window during the distribution process. Client i accepts service outside the time window, but there is a penalty cost for failing to provide service within the appointed time, and the total penalty cost PC equals to

 (13)

 : Time window when the customer i expects to be served 

Pe: Cost of waiting for the unit time if the vehicle arrives at customer node in advance (yuan/min)

Pl: Cost of punishing for the unit time if the vehicle is late to the customer node (yuan/min)

 : Time at which the kth vehicle arrives at point i

 : Time at which the kth vehicle leaves point i

Reviewer#2, Concern # 12: In the proposed model, it lacks the constraints for y_jkw and z_ijkw. It means that if z_ijkw=1, y_jkw must be 1. Whereas, if y_jkw=1, z_ijkw can be 1 or 0.

Author response: Thank you for your constructive and helpful suggestion. I have modified the constraints for y_jkw and z_ijkw in section 2.8:

 （22）

 （23）

Reviewer#2, Concern # 13: In the proposed model, constraints that vehicles must return the distribution center are omitted.

Author response: Thank you for your constructive and helpful suggestion. I have modified it as required:

 （24）

Eqn. (24) shows that all vehicles start and end at the distribution center. 

Reviewer#2, Concern # 14: In section 4.2.2, there is no figure 4.

Author response: Thank you for your constructive and helpful suggestion. I readjusted the diagram and normalized the number.

Reviewer#2, Concern # 15: The discussion for figure 2 is not appropriate because there is no temperature coordinate in figure 2.

Author response: Thank you for your constructive and helpful suggestion. I modified Figure 2 to Figure 3, and added a temperature annotation in Figure 3 to illustrate.

Reviewer#2, Concern # 16: Figure 5 should be Figure 4.

Author response: Thank you for your constructive and helpful suggestion. There is some error in figure numbering in the article, so I have modified the number of the Figure as required.

Reviewer#2, Concern # 17: is the optimal refrigerated temperature observed in figure 3. Why not set the controlled refrigerated temperature to in the section 4.2.3.

Author response: Thank you for your constructive and helpful suggestion. I have modified the number of the Figure as required. After modification, the optimal temperature control temperature is , so the following experiments in this paper are calculated according to this case( ).

Reviewer#2, Concern # 18: In this paper, several variables are not defined before they are used (e.g., d, a, v_c, v_f, ce and etc.) and some variable definitions are duplicated (e.g., ϑ and h, k and t). A lot of variables are higher than the text and not italic. The subscripts of some variable are wrong (e.g. TL, t_(i-1)i^a ). The authors should be checked all paper carefully and correct all mistakes

Author response: Thank you for your constructive and helpful suggestion. We updated the manuscript by declaring all variables in section 2.2, at the same time, the duplicated variables are removed such as TL, WL, w and so on. ϑ is the length of the route section, H is the route section number, and They have different meanings. 

Matrices are bolded and variables are italicized. We have unified the variable WL instead of TL, and We corrected the variable t_(i-1)i^a in section 2.4:

 (4)

I have rearrangement and corrected all the variables: Matrices are bolded, and variables are italicized.

---

## [Decision Letter · Decision Letter 1]

26 Jun 2020

Optimization of transportation routing problem for fresh food in time-varying road network: considering both food safety reliability and temperature control

PONE-D-20-08429R1

Dear Dr. Zhao,

We’re pleased to inform you that your manuscript has been judged scientifically suitable for publication and will be formally accepted for publication once it meets all outstanding technical requirements.

Kind regards,

Feng Chen

Academic Editor

PLOS ONE

Additional Editor Comments (optional):

Reviewers' comments:

Reviewer's Responses to Questions

**Comments to the Author**

1. If the authors have adequately addressed your comments raised in a previous round of review and you feel that this manuscript is now acceptable for publication, you may indicate that here to bypass the “Comments to the Author” section, enter your conflict of interest statement in the “Confidential to Editor” section, and submit your "Accept" recommendation.

Reviewer #1: All comments have been addressed

2. Is the manuscript technically sound, and do the data support the conclusions?

Reviewer #1: Yes

3. Has the statistical analysis been performed appropriately and rigorously? 

Reviewer #1: Yes

4. Have the authors made all data underlying the findings in their manuscript fully available?

Reviewer #1: Yes

5. Is the manuscript presented in an intelligible fashion and written in standard English?

Reviewer #1: Yes

6. Review Comments to the Author

Reviewer #1: (No Response)

7. PLOS authors have the option to publish the peer review history of their article (what does this mean?). If published, this will include your full peer review and any attached files.

Reviewer #1: No

---

## [Editor Report · Acceptance letter]

6 Jul 2020

PONE-D-20-08429R1 

Optimization of transportation routing problem for fresh food in time-varying road network: considering both food safety reliability and temperature control 

Dear Dr. Zhao:

I'm pleased to inform you that your manuscript has been deemed suitable for publication in PLOS ONE. Congratulations! Your manuscript is now with our production department. 

Kind regards, 

on behalf of

Dr. Feng Chen 

Academic Editor

PLOS ONE